# Impact of Abdominal Obesity on Frailty Development: A Web-Based Survey Using a Smartphone Health App

**DOI:** 10.3390/geriatrics10060147

**Published:** 2025-11-08

**Authors:** Hisayo Yokoyama

**Affiliations:** 1Research Center for Urban Health and Sports, Osaka Metropolitan University, 3-3-138 Sugimoto, Sumiyoshi-ku, Osaka-shi, Osaka 558-8585, Japan; yokoyama_hisayo@omu.ac.jp; Tel.: +81-06-6605-2947; 2Department of Environmental Physiology for Exercise, Osaka Metropolitan University Graduate School of Medicine, 3-3-138 Sugimoto, Sumiyoshi-ku, Osaka-shi, Osaka 558-8585, Japan

**Keywords:** habitual exercise, health app, older health, primary healthcare, web-based survey

## Abstract

**Background/Objectives**: Identifying adults at high risk of frailty and implementing appropriate interventions are critical for extending healthy life expectancy. This retrospective cohort study examined whether abdominal obesity predicts frailty progression over one year among 2962 community-dwelling adults aged 30–79 years in Osaka Prefecture, Japan. **Methods**: Data were collected from 2962 individuals (mean age, 62.7 ± 8.8 years) who completed annual surveys through a health application in both 2023 and 2024 and had available waist circumference data. Frailty was assessed using the *Kihon Checklist*. Logistic regression analysis was performed to identify predictors of frailty progression. **Results**: At baseline (2023), 23% of participants had abdominal obesity, and 18% were categorized as frail. Among 2431 participants who were non-frail at baseline, the incidence of frailty after one year was significantly higher among those with abdominal obesity than those without (10.5% vs. 7.2%, *p* = 0.011). However, in the multivariate logistic regression analysis, frailty awareness (“know well” vs. “do not know,” adjusted odds ratio [aOR] = 0.341, 95% confidence interval [CI] 0.212–0.548), regular exercise habits (aOR = 0.596, 95% CI 0.382–0.930), and prefrailty status (aOR = 1.767, 95% CI 1.602–1.950) were significant predictors of frailty development, whereas abdominal obesity was not independently associated with frailty progression after adjustment. **Conclusions**: Although abdominal obesity was associated with frailty onset in crude analyses, this association became non-significant after adjustment. Greater frailty awareness and regular exercise appear to reduce the risk of frailty development, suggesting that lifestyle education and public awareness initiatives may help mitigate the impact of abdominal obesity on frailty progression.

## 1. Introduction

Japan is currently facing an unprecedented super-aging society, with the baby boomer generation reaching 75 years or older by 2025. Consequently, medical and long-term care costs are expected to escalate, placing a heavy financial burden on the national healthcare system. Extending healthy life expectancy has therefore become a critical national goal. However, the gap between life expectancy and healthy life expectancy has remained unchanged for over two decades [1].

Frailty is a multidimensional syndrome characterized by declines in physical, psychological, and social functioning [2]. It is also a precursor condition for those who eventually require long-term care [3]. Early identification of individuals at high risk of frailty and timely interventions are essential for maintaining independence and improving health outcomes.

Although obesity is often considered the opposite of frailty, several of its underlying factors—such as reduced physical activity, poor dietary habits, and impaired muscle protein synthesis due to metabolic dysfunction [4]—can contribute to frailty development. Clarifying whether abdominal obesity independently increases the risk of frailty would aid in developing more targeted preventive strategies.

Therefore, this study aimed to examine the association between abdominal obesity and frailty progression among community-dwelling adults in Osaka Prefecture, Japan, and to test the hypothesis that abdominal obesity increases the risk of frailty incidence over one year.

## 2. Materials and Methods

### 2.1. Study Design and Participants

This retrospective observational cohort study was conducted among community-dwelling adults and is reported in accordance with the *STROBE* Statement. A completed *STROBE* checklist is provided as Appendix A.

Participants were community-dwelling individuals aged 30–79 years who responded to annual frailty questionnaires via the “ASMILE” healthcare smartphone application [5] in both 2023 and 2024. Individuals aged <30 or ≥80 years were excluded due to small sample sizes and, for those in their 20 s, the absence of abdominal obesity cases. This age range was selected to ensure representativeness and analytical stability for the target population.

Ethical approval for this study was obtained from the Research Ethics Committee of the Faculty of Liberal Arts, Sciences, and Global Education, Osaka Metropolitan University (Approval No. 2022-09; approved 1 November 2022). Informed consent was obtained from all participants. Specifically, residents who agreed to participate in the study consented to link their questionnaire responses with health checkup data registered in their ASMILE account by checking the designated consent box after reading the study explanation provided during ASMILE account registration.

### 2.2. Measures

This retrospective cohort study was conducted from February 2023 to February 2024. The survey included the *Kihon Checklist* (KCL) for frailty assessment, along with questions on exercise habits and frailty awareness. The KCL is a 25-item, yes/no questionnaire developed by the Japanese Ministry of Health, Labour, and Welfare to identify older adults at risk of requiring long-term care [6] (Appendix A). The total number of affirmative responses was used as the KCL score for each year. Following established criteria, a KCL score ≥ 7 was classified as frailty [7].

Participants were also asked about their exercise habits (“Do you exercise, such as walking, at least once a week?” [yes/no]) and frailty awareness (“Do you know the word frailty?” [do not know/have heard the word before/know a little/know well]). Self-reported height and weight (included in KCL item 12) were used to calculate body mass index (BMI).

All responses were linked to age and sex data registered in the ASMILE account. The application also automatically recorded daily step counts using the smartphone’s pedometer function. Step count data were collected over 19 days during the February 2023 survey period, and the mean daily step count was calculated. Because pedometer data were automatically integrated with the ASMILE system, no missing data occurred. However, days with <100 or >50,000 steps were excluded from the mean calculation, as such values were deemed implausible.

### 2.3. Waist Circumference

Waist circumference (WC) data from 2023 were obtained through clinician measurements conducted during Japan’s *Specific Health Checkups* (SHCs)—a national health screening program focusing on metabolic syndrome—or through annual employee health examinations conducted within the 10 months preceding the survey.

SHC data were managed by Osaka Prefecture and linked to survey responses by the Osaka Prefectural Government. Measurements from employee health examinations were voluntarily entered by participants into the ASMILE app. Using these procedures, WC data were securely linked with the questionnaire dataset. All data processing was conducted by Osaka Prefecture, and anonymized data were provided to the research team.

Abdominal obesity was defined as WC ≥ 85 cm for men and ≥90 cm for women, according to criteria established by the Japanese Committee for the Diagnostic Criteria of Metabolic Syndrome and endorsed by the Japan Society for the Study of Obesity (JASSO) [8]. These thresholds reflect body composition characteristics specific to Japanese adults. While the International Diabetes Federation (IDF) [9] recommends WC cutoffs of ≥90 cm for men and ≥80 cm for women for Asian populations, country-specific standards remain acceptable within the IDF framework. Therefore, this study applied the Japanese national criteria (≥85 cm for men and ≥90 cm for women), which are commonly used in health screening and epidemiological studies in Japan.

### 2.4. Statistical Analysis

The normality of all variables was assessed using the Kolmogorov–Smirnov test. Categorical variables were compared between groups using the chi-square test, except for frailty awareness, which was analyzed using the Cochran–Armitage test for trend (treating awareness as an ordinal variable). Continuous variables were compared using the unpaired *t*-test. Effect sizes were calculated using Cohen’s *d* and classified as small (0.20 ≤ *d* < 0.50), medium (0.50 ≤ *d* < 0.80), or large (*d* ≥ 0.80).

To identify predictors of frailty development, binary logistic regression analyses were conducted. A series of stepwise models (Models 1–5) was used to examine changes in the association between abdominal obesity and frailty onset after sequential adjustment for potential confounders and mediators:Model 1: Age, sexModel 2: Model 1 + abdominal obesityModel 3: Model 2 + baseline KCL scoreModel 4: Model 3 + exercise habits and frailty awarenessModel 5: Model 4 + mean daily steps

Explanatory variables included sex (1 = male, 2 = female), abdominal obesity (1 = yes, 0 = no), exercise habits (1 = yes, 0 = no), and frailty awareness (1 = do not know, 2 = have heard, 3 = know a little, 4 = know well). Mean daily steps were entered as a continuous variable per 1000-step increment. Prespecified interaction terms (age × abdominal obesity, sex × abdominal obesity, exercise habit × abdominal obesity) were also tested.

Results are presented as adjusted odds ratios (aORs) with 95% confidence intervals (CIs). Model fit and explanatory power were evaluated using the model χ^2^, −2 log likelihood, and Nagelkerke R^2^ statistics. Diagnostics included assessments for multicollinearity (variance inflation factor), linearity in the logit for continuous variables (Box–Tidwell procedure), and model calibration (Hosmer–Lemeshow test). As the study aimed to assess associations rather than predictive performance, the area under the receiver operating characteristic (ROC) curve was not calculated.

As a sensitivity analysis, changes in KCL scores from 2023 to 2024 were analyzed as continuous variables using linear regression to assess the robustness of the findings. A post hoc power analysis was also conducted using G*Power 3.1.9.7.

All statistical analyses were performed using SPSS version 27.0 (IBM Corp., Armonk, NY, USA). Statistical significance was set at *p* < 0.05.

## 3. Results

### 3.1. Participant Characteristics and Abdominal Obesity Prevalence

A flowchart illustrating participant selection and inclusion is presented in Figure 1. A total of 6857 adults aged 30–79 years participated in both the 2023 and 2024 surveys. Among them, WC data from the 2023 survey were available for 2962 respondents, who were included in the final analysis (mean age, 62.7 ± 8.8 years). A comparison between included (n = 2962) and excluded (n = 3895) respondents is presented in Appendix A.

Table 1 summarizes the prevalence of abdominal obesity among participants. Overall, 23% of participants had abdominal obesity, which was significantly more prevalent among men than women (41.2% vs. 10.2%, *p* < 0.001).

### 3.2. Baseline Differences Between Participants with and Without Abdominal Obesity

In the 2023 survey, participants with abdominal obesity were older than those without (64.8 ± 8.1 vs. 62.1 ± 9.0 years, *p* < 0.001; Cohen’s *d* = 0.327). They also had higher BMI (25.5 ± 2.9 vs. 20.9 ± 2.3 kg/m^2^, *p* < 0.001; *d* = 1.842) and WC (92.4 ± 6.7 vs. 76.8 ± 6.7 cm, *p* < 0.001; *d* = 2.313). Frailty prevalence was also higher among those with abdominal obesity (21.5% vs. 16.8%, *p* = 0.006). Conversely, frailty awareness was significantly lower among participants with abdominal obesity compared with those without (do not know/have heard/know a little/know well: 19.4%/22.8%/29.6%/28.2% vs. 14.0%/18.6%/32.6%/34.8%, Cochran–Armitage trend test = 22.5, *p* < 0.001). Mean daily step counts and exercise habits did not differ significantly between groups.

### 3.3. Incidence of Frailty by Obesity Status and Related Factors

Among 2431 participants who were non-frail in 2023, 7.9% developed frailty within one year. As shown in Figure 2, the incidence of frailty was higher in participants with abdominal obesity than in those without (10.5% vs. 7.2%, *p* = 0.011). Participants who developed frailty were less likely to engage in regular exercise than those who did not (80.3% vs. 90.4%, *p* < 0.001) and had lower frailty awareness (do not know/have heard/know a little/know well: 28.0%/26.9%/24.9%/20.2% vs. 12.3%/17.6%/32.5%/37.6%, Cochran–Armitage trend test = 55.3, *p* < 0.001).

### 3.4. Multivariable Analysis of Predictors for Frailty Onset

Multivariable logistic regression results are presented in Table 2. Higher baseline KCL scores were consistently and strongly associated with frailty development across all models (aOR ≈ 1.8 per 1-point increase, *p* < 0.001). Abdominal obesity was not independently associated with frailty progression in any model except Model 2 (Model 5: aOR = 1.117, 95% CI 0.767–1.627, *p* = 0.566). In contrast, exercise habits (aOR ≈ 0.6, *p* = 0.023) and greater frailty awareness (aOR = 0.34–0.36 for higher categories, *p* < 0.001) emerged as significant protective factors. The inclusion of lifestyle-related variables (Models 4–5) improved model fit (Nagelkerke R^2^ ≈ 0.20).

### 3.5. Interaction Between Abdominal Obesity and Exercise Habit

A significant interaction was identified between abdominal obesity and exercise habit (*p* for interaction = 0.030). Stratified logistic regression analyses based on exercise habits (Appendix A) revealed that although the simple effects were not statistically significant within either subgroup, the direction of associations differed between those with and without exercise habits. These findings suggest that abdominal obesity may influence frailty development primarily among individuals who engage in regular exercise.

### 3.6. Sensitivity Analysis and Model Robustness

No evidence of multicollinearity was observed (all variance inflation factors < 1.4), and the assumption of linearity in the logit was satisfied. The model demonstrated good calibration (Hosmer–Lemeshow *p* = 0.539). Model stability was confirmed based on events per variable (EPV). With 193 incident frailty cases and seven explanatory variables in the fully adjusted model, the EPV was 27.6—exceeding the conventional threshold of 10—indicating adequate reliability.

In the sensitivity analysis using changes in KCL scores from 2023 to 2024 as a continuous outcome, abdominal obesity was not significantly associated with score increases (β = 0.186, 95% CI −0.020 to 0.393, *p* = 0.077), consistent with the primary findings. Based on the observed incidence and sample size, the detectable effect size for abdominal obesity corresponded to an odds ratio of approximately 1.5 with 70% statistical power.

## 4. Discussion

In this study, the proportion of participants who progressed to frailty over one year was slightly but significantly higher among those with abdominal obesity than among those without. Therefore, individuals with abdominal obesity should be prioritized for frailty prevention interventions. However, logistic regression analysis indicated that low frailty awareness, lack of exercise habits, and prefrailty status independently predicted frailty development, whereas abdominal obesity was not a significant explanatory factor. These findings suggest that, over a short follow-up period of one year, the adverse effects of abdominal obesity on frailty progression may be mitigated if individuals possess sufficient knowledge of frailty and engage in positive behavioral changes, such as adopting regular exercise. Thus, individuals with abdominal obesity should not only be encouraged to manage weight through an appropriate diet but also be educated about frailty and supported in increasing physical activity to address this growing public health concern.

A previous prospective cohort study demonstrated that visceral obesity and insulin resistance are associated with frailty development [10], and the results of this study are consistent with its findings. As visceral fat accumulates, immune cells such as macrophages infiltrate adipose tissue, inducing chronic low-grade inflammation [11]. Cytokines released into the circulation subsequently promote muscle protein degradation and muscle fiber loss, while inhibiting muscle stem cell differentiation [11,12]. Insulin resistance driven by visceral adiposity also contributes to muscle degeneration. Elevated serum free fatty acids (FFAs), caused by excessive lipolysis under systemic insulin resistance, result in ectopic fat deposition in skeletal muscle cells [11,12], disrupting insulin’s anabolic effects on muscle protein turnover [13]. This vicious cycle of visceral adiposity, insulin resistance, and skeletal muscle atrophy leads to sarcopenic obesity—a distinct metabolic condition [14].

Physical activity has both preventive and ameliorative effects on each component of this vicious cycle, with observable benefits within a few months [15,16]. Previous studies have also shown that exercise interventions can prevent or delay frailty progression in older adults [17]. This may explain why regular exercise habits were a significant protective factor against frailty development in our study. Moreover, higher frailty awareness was independently associated with reduced frailty risk. Providing health education and promoting public awareness can empower individuals with self-care knowledge and skills, thereby fostering disease prevention [18]. In other words, enhanced health literacy may lower frailty risk by encouraging healthier behaviors, such as better dietary and exercise management, effective communication with healthcare providers, and routine health checkups. These insights can guide local governments in developing effective health promotion policies.

Cohort studies have consistently shown that prefrailty is a strong predictor of frailty onset in community-dwelling older adults [19]. A recent report by Lin et al. found that prefrail status increased frailty risk even within one year [20]. Our finding that prefrailty status had a substantial confounding effect on frailty development aligns with these studies. Interestingly, the effect of abdominal obesity became nonsignificant after adjusting for prefrailty status, indicating that the crude association was largely explained by pre-existing declines in physical or psychosocial function measured by the Kihon Checklist (KCL). In other words, individuals with abdominal obesity may already exhibit prefrail characteristics that predispose them to frailty. As discussed, abdominal obesity may represent an early manifestation of frailty through sarcopenic pathways, reflected in higher KCL scores. Therefore, interventions targeting visceral fat accumulation may be particularly critical during the prefrailty stage.

Unexpectedly, the interaction term suggested that the effect of abdominal obesity on frailty development was stronger among participants with exercise habits, despite the known protective effects of exercise. Additionally, mean daily steps were not an independent predictor of frailty onset. These findings may reflect variability in exercise type and intensity, which were not evaluated in this study. It is also possible that individuals with early functional decline had already initiated exercise following health advice. Future research should explore the influence of exercise type, duration, and intensity to clarify this relationship.

According to the 2019 National Health and Nutrition Survey, the prevalence of abdominal obesity was 45% among males aged 30–39 years and approximately 60% among those in their 40 s to 70 s. Among females, the prevalence increased with age, reaching about 25% among those in their 60 s to 70 s [21]. In our study, the prevalence of abdominal obesity was higher among males than females, although both were lower than national averages. This may be due to age distribution bias and participant characteristics, as the sample comprised individuals who used the ASMILE health app, self-recorded checkup results, or linked their specific health check (SHC) data. Consequently, participants were likely more health-conscious than the general population, and the findings should be interpreted with caution. Further research is warranted to determine whether abdominal obesity similarly influences frailty development in populations with higher obesity prevalence.

Several limitations should be acknowledged. First, this study used retrospectively collected data from a smartphone-based health app and existing health checkup records. Thus, the temporal relationship between exposure and outcome cannot be definitively established, and causal inference should be made cautiously. Second, the study population was restricted to adults aged 30–79 years who provided valid data in two consecutive years, excluding individuals without WC data and younger or older adults. While this design improved comparability, selection bias cannot be ruled out. Excluded individuals tended to be younger, more likely female, and exhibited higher KCL scores and lower exercise rates, suggesting that the included sample may have had better health status, potentially attenuating associations and limiting generalizability. Third, frailty was assessed using the KCL, originally validated for adults aged ≥65 years, though it is widely used for those aged ≥40 years in national programs. Fourth, abdominal obesity data were derived from both SHCs and annual employee health examinations, which could not be completely distinguished. This may have introduced selection or confounding bias, as SHCs primarily target self-employed or unemployed individuals whose socioeconomic status may differ from those undergoing employee-based examinations. Finally, statistical power was moderate (~70%) given the 8% event rate and sample size, possibly limiting the detection of minor effects. Despite these limitations, this study provides valuable population-level evidence linking abdominal obesity, exercise habits, and frailty awareness among community-dwelling adults, emphasizing modifiable factors for early frailty prevention.

## 5. Conclusions

The findings of this study highlight the importance of identifying appropriate targets for intervention to prevent frailty progression through simple and noninvasive assessments, such as measuring WC, and implementing proactive health guidance. Moreover, increasing knowledge and awareness about frailty may promote behavioral changes that help prevent its onset within a short period, such as one year. Therefore, enhancing frailty awareness and education across different age groups represents an urgent public health priority that requires immediate attention.

## Figures and Tables

**Figure 1 geriatrics-10-00147-f001:**
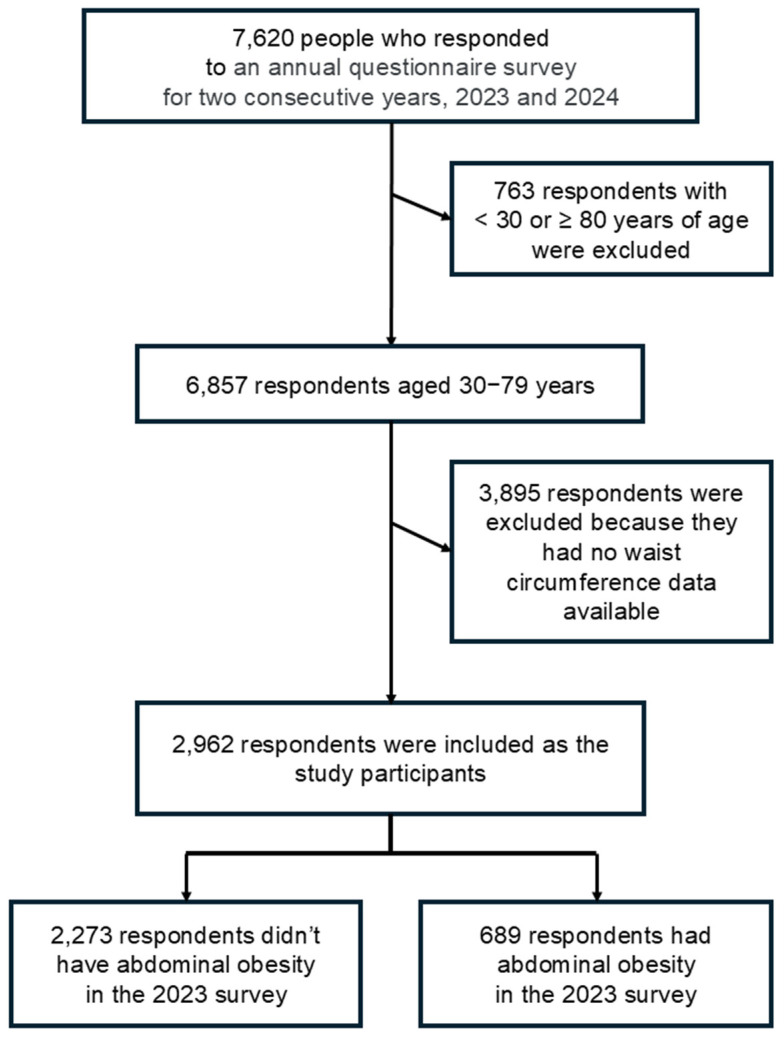
Flow chart from the participants’ selection to analysis.

**Figure 2 geriatrics-10-00147-f002:**
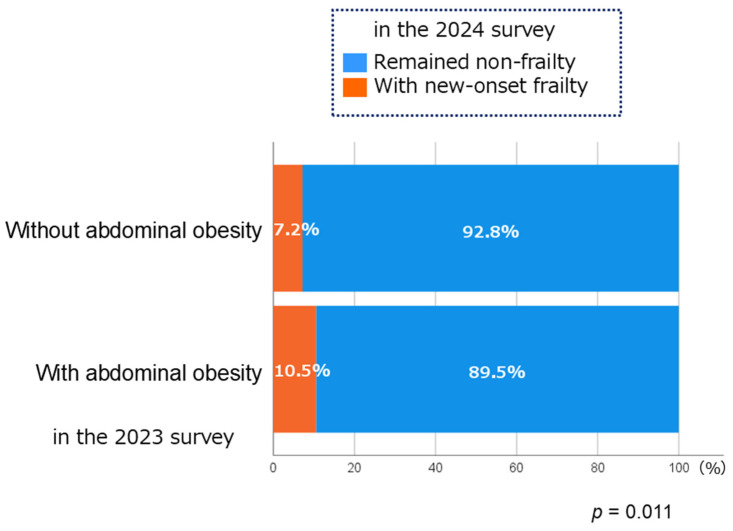
Percentages of participants with new-onset frailty in the 2024 survey.

**Table 1 geriatrics-10-00147-t001:** Prevalence of abdominal obesity among the participants.

	Male			Female		
	n = 1250			n = 1712		
		abdominal obesity			abdominal obesity	
	n	no (%)	yes (%)	n	no (%)	yes (%)
30–39 years	10	80.0	20.0	17	100.0	0.0
40–49 years	75	72.0	28.0	159	93.7	6.3
50–59 years	252	62.3	37.7	450	90.0	10.0
60–69 years	488	57.8	42.2	734	90.7	9.3
70–79 years	425	55.1	44.9	352	85.5	14.5

**Table 2 geriatrics-10-00147-t002:** Factors associated with frailty onset based on logistic regression analysis with stepwise models.

	Model 1	Model 2	Model 3	Model 4	Model 5
Variables	age + sex	Model 1 + Abdominal obesity	Model 2 + KCL score	Model 3 + Exercise habit + Frailty awareness	Model 4 + Mean daily steps
Age (per year)	0.995 (0.979–1.011) *p* = 0.523	0.993 (0.977–1.010) *p* = 0.428	0.997 (0.980–1.015) *p* = 0.779	1.010 (0.992–1.030) *p* = 0.276	1.011 (0.992–1.030) *p* = 0.262
Female (vs. male)	0.729 (0.543–0.980) *p* = 0.036	0.819 (0.595–1.128) *p* = 0.221	0.970 (0.693–1.357) *p* = 0.859	1.147 (0.808–1.630) *p* = 0.443	1.169 (0.812–1.683) *p* = 0.402
Abdominal obesity (yes)		1.425 (1.003–2.023) *p* = 0.048	1.097 (0.757–1.589) *p* = 0.624	1.107 (0.762–1.608) *p* = 0.594	1.117 (0.767–1.627) *p* = 0.566
KCL score (per point)			1.815 (1.648–1.998) *p* < 0.001	1.767 (1.602–1.950) *p* < 0.001	1.767 (1.602–1.950) *p* < 0.001
Frailty awareness (vs. “do not know”)					
“have heard the word before”				0.678 (0.436–1.054) *p* = 0.085	0.681 (0.438–1.060) *p* = 0.088
“know a little”				0.363 (0.233–0.567) *p* < 0.001	0.364 (0.233–0.569) *p* < 0.001
“know well”				0.341 (0.212–0.548) *p* < 0.001	0.341 (0.212–0.548) *p* < 0.001
Exercise habit (yes)				0.611 (0.399–0.935) *p* = 0.023	0.596 (0.382–0.930) *p* = 0.023
Mean daily steps (per 1000 step)					1.007 (0.972–1.043) *p* = 0.710
Model χ^2^ (df)	4.465 (2), *p* < 0.107	8.278 (3), *p* < 0.041	186.90 (4), *p* < 0.001	221.84 (8), *p* < 0.001	221.98 (9), *p* < 0.001
−2 Log Likelihood	1366.55	1362.74	1161.24	1126.29	1126.16
Nagelkerke R^2^	0.004	0.008	0.174	0.205	0.205

Values are adjusted odds ratios (95% confidence intervals) from binary logistic regression; KCL score, the number of applicable items in Kihon Checklist/25 items; df, degrees of freedom.

## Data Availability

The data supporting the findings of this study are openly available on FigShare at https://doi.org/10.6084/m9.figshare.28253060.v1 (accessed on 21 October 2025).

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
