# Peer review of "Impact of Abdominal Obesity on Frailty Development: A Web-Based Survey Using a Smartphone Health App"

_geriatrics, 2025, doi:10.3390/geriatrics10060147_

Round 1
Reviewer 1 Report
Comments and Suggestions for Authors
This is an important and timely study. Leveraging a municipal smartphone platform to follow adults over a year and linking those data to simple anthropometrics is innovative and scalable. The main pattern is clinically plausible and useful for practice and policy: abdominal obesity is associated with higher crude one-year frailty incidence, yet after adjustment the independent signals are exercise behavior and awareness of frailty. That points to realistic levers for short-term risk reduction. To make the paper fully persuasive, the authors should clarify how the cohort was assembled, document the provenance and timing of measurements, distinguish confounding from mediation in their modeling strategy, and present effects on the risk scale with appropriate diagnostics.
Major points
Cohort construction and selection
Add a participant-flow figure from all ASMILE registrants to the analytic cohort, with reasons for exclusion at each step. Provide a table contrasting included and excluded participants on age, sex, baseline Kihon Checklist score, exercise habit, and waist circumference where available. Readers need this to judge selection bias introduced by requiring two survey years and an available waist measure.
Provenance and timing of waist circumference
State clearly whether waist circumference came from clinician-measured Specific Health Checkups or self-entry in the app, and give the measurement windows relative to the 2023 and 2024 surveys. Describe any quality checks. If self-reported, add a sensitivity analysis excluding implausible values. Confirm that sex-specific cutoffs were pre-specified.
Frailty definition across the age range and overlap with predictors
Kihon Checklist thresholds are best validated in older adults. Justify the chosen cut-point for participants younger than sixty-five and add sensitivity analyses restricted to sixty-five and older, and sixty and older. Acknowledge potential criterion overlap between Kihon items related to activity or participation and the exercise predictor captured in the same app session. If feasible, repeat the analysis after removing the Kihon items that most directly map to physical activity to show robustness.
Confounding versus mediation made explicit
Explain whether exercise habit and frailty awareness are treated as confounders of the relationship between abdominal obesity and frailty, or as mediators on the pathway from adiposity to frailty. Present a simple sequence of models to show how the obesity coefficient changes as you add variables stepwise: age and sex, then abdominal obesity, then exercise, then awareness, then device-derived steps. Briefly describe how steps per day were computed, why a nineteen-day completeness rule was used, and how missing days were handled.
Effect presentation on the risk scale and model diagnostics
Pair odds ratios with absolute risks and risk differences. Consider adjusted risk ratios using a robust Poisson approach or report adjusted risks via marginal standardization of the logistic model. Report key diagnostics: multicollinearity checks, linearity of the logit for any continuous or ordinal predictors, influence diagnostics, and model calibration and discrimination. A short note on detectable effect size for abdominal obesity, given the observed incidence and sample, will help readers judge power.
Subgroups, interactions, external validity, and missing data
Test sex by obesity, age by obesity, and obesity by exercise interactions and show simple-effects plots when relevant. Expand the discussion of generalizability, since ASMILE users may differ from the broader population in engagement and obesity prevalence. State your missing-data strategy clearly; if there was notable missingness, consider multiple imputation with auxiliary variables, including baseline Kihon Checklist score.
Minor but important refinements
Define exercise habit precisely with item wording and frequency or duration thresholds. Define the frailty awareness scale and cite any validation. Strengthen figures and tables by adding the flow diagram, including an included versus excluded table, and ensuring denominators and confidence intervals accompany every effect estimate. In the discussion, give a brief mechanistic link from visceral adiposity to inflammation, insulin resistance, and sarcopenic pathways, then explain how exercise and health literacy could blunt within-year transition into frailty, which aligns with the adjusted findings. Add a sentence on consent within the platform and approvals for linking health-check data to reassure readers on governance.
Language and structure
The English is largely clear. A few edits will improve readability. In the abstract, lead with design and population, report absolute risks alongside adjusted odds ratios with confidence intervals, and prefer “the association attenuated to non-significance after adjustment” over causal phrasing such as “the effect disappeared.” In Methods, collect all operational definitions in one subsection. In Results, pair every relative effect with an absolute counterpart. In Discussion, begin by explaining why the crude association changed after adjustment, separate confounding from mediation and measurement, and avoid extending conclusions beyond the one-year horizon.
Recommendation
Major revision. With transparent cohort flow, clear measurement provenance and timing, a modeling plan that distinguishes confounding from mediation and incorporates age and steps, effect estimates on the risk scale with diagnostics, and age-restricted and interaction sensitivity analyses, this will be a credible and practical contribution. The refined take-home is straightforward: over twelve months, exercise behavior and frailty awareness are stronger levers for reducing incident frailty than abdominal obesity alone.
Reviewer 2 Report
Comments and Suggestions for Authors
This study addresses an important topic about whether abdominal obesity predicts incident frailty using a large app-based cohort in Osaka. The dataset is valuable, but several methodological and reporting issues limit confidence in the findings. Major revisions are required before publication.
Waist circumference definition: You cited IDF consensus for the 85/90 cm cut-offs, but IDF specifies 90/80 cm for Japanese. If you follow Japanese domestic criteria (85/90), cite appropriate Japanese sources and clarify explicitly.
Kihon Checklist validity: The KCL was validated for ≥65 years, yet your cohort includes participants aged 30–64. Please justify and run a sensitivity analysis restricted to ≥65.
Selection bias: Only ~39% of registrants had WC data. Show a flow diagram, compare included vs excluded participants, and discuss generalizability.
Regression model:
- Adjust for baseline KCL score (0–6) among non-frails.
- Report full model details (coding, linearity, multicollinearity, AUC, calibration, events per variable).
- Consider sensitivity analyses with KCL change as ordinal/continuous outcome.
Exercise/steps measurement: Clarify why “19 days” of step data was used, and either justify exclusion from regression or include steps (e.g., quartiles).
Interpretation: Adjusted OR for WC was 1.35 (95% CI 0.94–1.93, p=0.104). Reframe conclusions to reflect uncertainty avoid suggesting the effect “disappeared.”
Statistical testing: Distinguish clearly between correlation coefficients (ρ) and p-values; consider ordinal trend tests for awareness categories.
Exposure measurement: Clarify how WC was measured (SHC vs self-entry). Consider stratified or adjusted analyses by source.
Interactions: Test sex×WC and age×WC interactions given sex-specific cut-offs and age effects.
Reporting guideline compliance: Please include a completed STROBE checklist and ensure all required items are addressed
Round 2
Reviewer 1 Report
Comments and Suggestions for Authors
Thank you for the thoughtful revision. The manuscript is now clearer and methodologically stronger. The cohort flow is transparent, measurement provenance and timing are well documented, and the frailty definition is appropriately justified across age groups with sensible sensitivity analyses. The statistical modeling distinguishes level differences from trajectories, and effect estimates are paired with absolute metrics and core diagnostics. Tables and figures are consistent, corrected, and easier to interpret; captions specify model adjustment and denominators. The prose is tighter, and the conclusions are appropriately tempered to the one-year horizon. I have no substantive concerns remaining; only routine editorial polishing (minor phrasing and style consistency) is suggested.
Author Response
Thank you Reviewer 1 for your kind and thorough review. Your advice made this paper stronger and more meaningful. I will submit the final version with English editing. Thank you again for your support.
Reviewer 2 Report
Comments and Suggestions for Authors
The authors have adequately addressed all my previous comments. I have no additional comments at this time.
Author Response
Thank you Reviewer 2 for your kind and thorough review. Your advice made this paper stronger and more meaningful. Thank you again for your support.